nanotechnology/biomaterials/materials science

protein-based biopolymers, carbohydrate-based biopolymers, green Ag nanoparticles, biological active biopolymers, effective anticancer biopolymer–Ag complex

**Author for correspondence:**
Altaf H. Basta
e-mail: altaf_basta2004@yahoo.com,
Altaf_Halim@yahoo.com

This article has been edited by the Royal Society of Chemistry, including the commissioning, peer review process and editorial aspects up to the point of acceptance.

# Synthesis and evaluation of protein-based biopolymer in production of silver nanoparticles as bioactive compound versus carbohydrates-based biopolymers

Altaf H. Basta[1], Vivian F. Lotfy[1], Khaled Mahmoud[2] and Nayera A. M. Abdelwahed[3]

[1]Cellulose and Paper Department, [2]Pharmacognosy Department, and [3]Chemistry of Natural and Microbial Products Department, National Research Centre, El-Buhouth Street, Dokki-12622 Cairo, Egypt

AHB, 0000-0003-1876-4378

This overall process deals with evaluating the performance of silver nanoparticles, synthesized from sodium caseinate (SC) as green biological active agent, in comparison with widely produced from carboxymethyl cellulose, other carbohydrates (oxidized nanocellulose fibres (OC) and starch (St)). The TGA, FTIR and TEM, as well as its antimicrobial activities toward pathogenic Gram-positive and Gram-negative bacteria in addition to the yeast strain *Candida albicans* NRRL Y-477 were examined. In addition, with regard to their anti-tumour activity, the evaluation was studied via many cancer cell lines against RPE1 (normal retina cell line). The results revealed that the SC–Ag(I) and CMC–Ag(I) complexes were formed in six- and five-membered chelate rings, respectively, as nanoparticles, while linear chelation structure was formed in case of OC–Ag(I) and St–Ag(I) complexes. The complexation of SC with Ag(I) ions was recommended as promising stable and antimicrobial agent, with lower free Ag(I) ions and particle size than other Ag-complexes. Moreover, it provided anti-tumour activity of most tested cell lines (*in vitro*), with the following sequence HCT116 > PC3 > HePG 2 > MCF-7 > A549 with IC$_{50}$ and IC$_{90}$ values of 25.8 and 54.73 µg ml$^{-1}$, 45.1 and 66.7 µg ml$^{-1}$, 64.3 and 110.7 µgml$^{-1}$, 71.4 and 114.8 µgml$^{-1}$

and 80.1 and 127.7 $\mu g\,ml^{-1}$, respectively. The promising effect of SC–Ag complex was also clear from its selective index versus RPE1 (normal retina cell line).

# 1. Introduction

Biopolymers are regarded as important natural materials used in many applications, such as functional paper, metal adsorbents and HCHO-free adhesives, as well as hydrogels for reclamation of sandy-calcareous soils, and medical applications for drug delivery systems [1–9]. Functional groups included biopolymers such as, carboxylate and amino groups that promote their complexation with metal ions for the use of semiconductors, magnetic and durable paper sheets. Moreover, they are used in medical applications as modern therapeutic, diagnostic and radiopharmaceutical agents, antimicrobial agents, anti-ulcer treatments, anti-arthritic agents, magnetic resonance imaging (MRI) contrast agents, and as radiopharmaceutical agents [10–15].

Many years ago, was silver regarded as the unique metal of valuable uses in treating and preventing the diseases. Its complexes were applied as biocide and/or biostatic [16], while its salts succeeded in treating some infections appearing resistant to modern antibiotics and drops of silver nitrate that are placed in newborn's eyes at birth for the prevision of contracting gonorrhea from mother [17]. Synthesizing of green silver nanoparticles gained the attention by many researchers due to its biological activity, due to its ability to anchor the microorganism cell wall, followed by penetrating, damaging of intracellular structures and biomolecules (protein, lipids and DNA). The other effects that were discussed include the release of silver nanoparticles (AgNPs), silver ions, cell membrane damage, AgNPs induced cellular toxicity, and oxidative stress caused due to the formation of free radical and reactive oxygen species (ROS). AgNPs are able to modulate the immune system of the cells by orchestrating inflammatory response, and consequently aid in the inhibition of microorganisms [18–20]. A green synthesis of AgNPs from plant extracts and its antimicrobial mechanisms were studied in detail by Oves *et al.* [21–23] and Durán *et al.* [24]

Green synthesizing and stabilizing of metallic nanoparticles (Ag, Au, etc.) were investigated by many researchers via the use of carbohydrate ethers (carboxymethyl cellulose, carboxymethyl starch and carboxymethyl chitosan) [25–27]. While, the efficiency of sodium caseinate, as a protein origin on synthesizing AgNPs as bioactive compound for inhibiting both microorganisms and anti-tumour versus carbohydrates (carboxymethyl cellulose, oxidized cellulose nano-fibrous and native starch), was the aim of this work. This was not studied previously. Therefore, the goal of this article was not only to provide AgNPs by new biopolymer, but also for optimizing the highly effective bioactive compound for biological and anti-tumour activities, together with lowering free Ag ions, and consequently the drawbacks from leaching it to another part of the patient's body.

# 2. Material and methods

## 2.1. Materials

— Protein-based biopolymer: Sodium caseinate (SC) was supplied from S.d.fine-chem Ltd.
— Cellulose-based biopolymers: Carboxymethyl cellulose sodium salt (CMC) was analytical-grade reagent purchased from Piochem; DS ~ 0.79. The oxidized cellulose (OC) used in this study was obtained from cotton linter oxidation by ammonium persulphate for 8 h according to Filipova *et al.* [28]. The degree of oxidation of the obtained product was $0.026\,mol\,mol^{-1}$ using conductometric titration [29].
— Starch (St) was kindly supplied from Starch and Glucose Company.
— Silver nitrate was purchased from Sigma-Aldrich.

## 2.2. Synthesis of polymer–Ag nanoparticles complexes

The synthesis method was performed by dissolving or suspended a definite weight of protein- or carbohydrate-based biopolymer (sodium caseinate (SC), carboxymethyle cellulose (CMC), oxidized cellulose fibres (OC) and starch (St)) in distilled water; then adding 6% $AgNO_3$ drop wise. After dissolution, the pH of the solution was adjusted to 12.5 under stirring for 60 min at 70°C. At the end of the reaction time, the solution acquired a yellowish colour indicating the formation of AgNPs. After separating the solution, the complex product was washed with methanol, filtered and dried in oven at

50°C. The amount of silver ions was detected by UV–vis absorption at 380 nm, according to the method reported previously [30].

## 2.3. Evidence of biopolymer–Ag complexes formation

— Estimation of reacted silver ions was determined by UV–vis analysis according to [30].
— FTIR spectra: Infrared spectra were recorded with a Jasco FT/IR, Nicolet and Model 670. The samples were mixed with KBr and pressed as tablets. The technique of Nelson & O'Connor [31] was used to calculate the crystallinity index (Cr.I). The mean strength of hydrogen bonds (MHBS) was calculated according to Levdik *et al.* [32].
— Thermogravimetric analysis (TGA): The non-isothermal TGA of the synthesized Ag-complexes in nanoparticle form, from different biopolymers was carried out by using Instrument SDT Q600 V20 Build 20 module (USA), under nitrogen atmosphere at a heating rate of 10°C min$^{-1}$ and temperature range from approximately 30°C to approximately 1000°C. The calculations of TGA measurements were carried out according to [33,34].
— Transmission electron microscopy (TEM): Morphological characterization of the synthesized complexes (as AgNPs) was carried out by using transmission electron microscopy, of type QUANTA FEG250, Japan (system running at 200 keV).

## 2.4. Antimicrobial activity

### 2.4.1. Agar diffusion method

The antibacterial activities of the synthesized compounds were tested against *Escherichia coli* NRRL B-210 and *Pseudomonas* NRRL B-23 (Gram-negative bacteria), *Bacillus subtilis* NRRL B-543 and *Staphylococcus aureus* NRRL B-313 (Gram-positive bacteria) using nutrient agar medium. The antifungal activity of these compounds was also tested against *Candida albicans* NRRL Y-477 using Sabouraud dextrose agar medium.

The synthesized AgNPs from protein- and carbohydrate-based biopolymers were screened *in vitro* for their antimicrobial activity against the aforementioned selected pathogenic microorganisms by agar diffusion method [35]. Suspension of 0.5 ml from each strain was added to a sterile nutrient agar medium at 45°C and the mixture was transferred to sterile Petri dishes and allowed to solidify. Holes of 6 mm in diameter were made using a cork borer. Amounts of 0.1 ml of the synthesized nano silver compounds from a stock solution of 20 µg ml$^{-1}$ were poured inside the holes. A hole filled with DMSO was also used as control. The plates were left for 1 h at room temperature as a period of pre-incubation diffusion. The same method was carried out using Sabouraud dextrose agar medium using *C. albicans* NRRL Y-477. The plates were then incubated at 35°C for 24 h and observed. The diameters of inhibition zone were measured and compared with that of the standard antibiotics ciprofloxacin (5 µg ml$^{-1}$) and nystatin (100 units ml$^{-1}$) for antibacterial and antifungal activity, respectively.

### 2.4.2. Minimum inhibitory concentration

The microdilution broth method was used to obtain the minimum inhibitory concentration (MIC) of the synthesized AgNPs (biopolymer–Ag complexes) [36] against the previously mentioned standard strains using nutrient and Sabouraud dextrose broth for antibacterial and antifungal assay, respectively. A loop full from each strain suspension was inoculated in 5 ml of sterilized broth media in test tubes and incubated at 35°C for 24 h. The samples were prepared by weighting and dissolving in a minimal volume of DMSO forming a stock solution of 20 µg ml$^{-1}$ and were serially diluted in sterile media broth in the range of 20 to 1 µg ml$^{-1}$. A 50 µl from the 24 h strains cultures ($10^8$ CFU ml$^{-1}$) was added to each tube containing the serially diluted synthesized nano silver compounds as well as a sterility control and a growth control containing culture broth plus DMSO and without antimicrobial substance. Test tubes were incubated at 35°C for 24 h. The growth of the tested microorganisms was determined by measuring the turbidity using Agilent Technologies Cary 100 series UV–vis spectrophotometer at 600 nm after 24 h. Thus, the MIC was generally read as the smallest concentration of synthesized nano silver compounds in the series that prevents the development of visible growth of the test organism. All of the experiments were done in duplicate.

## 2.5. Anti-cancer activity MTT assay (*in vitro* bioassay)

The complexes of Ag(I) with different biopolymers were tested as anti-cancer agents to treat five human tumour cell lines, *in vitro* (colon HCT116, lung carcinoma A549, human hepatocellular carcinoma HePG 2, prostate PC3 and human caucasian breast adenocarcinoma MCF7). Its inhibition efficiency were assessed against RPE1 (normal retina cell line), both via using MTT bioassays [37–39]. It depends on the reduction of yellow MTT (3-(4,5-dimethylthiazol-2-yl)-2,5-diphenyl tetrazolium bromide) to purple formazan.

The concentrations examined of complexes were 100, 50, 25, 12.5, 6.25, 3.125, 1.56 and 0.78 µg ml$^{-1}$, and a probit analysis, using SPSS 11 program, was carried for estimating the concentrations which provide inhibition 50% and 90% of cancer cells in 48 h.

# 3. Results and discussion

## 3.1. Characterization of protein-based AgNPs versus carbohydrate-based AgNPs

### 3.1.1. FTIR spectra

The changes in the structure of protein- and carbohydrate-based biopolymers due to chelation with Ag(I) ions together with specifying which functional groups are able to share as chelating sides were studied via FTIR-spectra and are illustrated in figure 1. The IR measurements (mean strength of hydrogen bonds (MSHB) and crystallinity index (Cr.I)), are recorded in table 1. With regard to IR spectra of protein-based biopolymer (SC) and SC–Ag(I) complex, it is clear that, SC has the characterized bands at 3448, 2962 and 2106 cm$^{-1}$. These bands are related to asymmetrical and symmetrical stretching vibrations of OH and/or NH, and CH groups. The bands at 1617 and 1519 cm$^{-1}$ are assigned to C=O groups of amide I, amide II and carboxylate groups; while C–O of carboxylate appeared at 1468 cm$^{-1}$. The band of CH–OH bending vibration is observed at 1129–1136 cm.$^{-1}$ Due to chelation with Ag(I) ions, the red shift of bands related to stretching vibration of OH/NH and C=O are observed, whereas the peak of OH/NH band is shifted from 3448 to 3432 cm$^{-1}$ ($\Delta v = 16$ cm$^{-1}$); and C=O from 1643 to 1633 cm$^{-1}$ ($\Delta v = 10$ cm$^{-1}$) with the disappearance of band at 1519 cm$^{-1}$. The band at 1458 cm$^{-1}$ is shifted to 1399 cm$^{-1}$. This observation realized the chelation occurred via NH and COO$^-$ groups including the caseinate, is more possible, with the formation six-member ring chelation geometry (figure 2). Increasing the intensity of bands at range 840–599 cm$^{-1}$ is related to Ag–O bond formation. The presence of more functional groups as chelating sides is ascribed the greatest Ag(I) content (table 1)

In case of CMC, the position of OH stretching vibration of band on complexation is slightly shifted to higher wavenumber (from 3425 to 3428 cm$^{-1}$); while the bands related to C=O of COO$^-$ are decreased with the disappearance of band related to C–O (1419 cm$^{-1}$). This observation is evidence the OH groups did not include the chelation, but the chelation sides are via COO$^-$ and ether linkage of 1ry alcohol, with the formation five-member ring chelation geometry (figure 2). Moreover, some variation in the shape of the band related to the stretching vibration of OH group is noted, whereas the broadness in OH band of unchelated CMC is due to the formation of different strength of hydrogen bond (at range 4000–3000 cm$^{-1}$). The decrease of this broadness due to complexation with increasing the values of MSHB and Cr.I gives us the information dealing with the formation of stronger hydrogen bonds via free OH together with covalent bonds and ionic bonds between carboxylate and Ag ions. For this reason, the MSHB and Cr.I are increased from 1.85 to 6.04, and from 0.55 to 2.13, respectively.

For the case of COO-containing cellulose derivative (oxidized cellulose nano fibres; OC) table 1 and figure 1 show slight shift of bands related to OH of OC biopolymer due to chelation (from 3414 to 3419 cm$^{-1}$). This indicates the chelation is performed via ionic bonds with liberation of the OH including intra and inter hydrogen bonds. Due to the stronger ionic interaction of COO$^-$ of OC with silver ions than OH there is an increase in its Cr.I. The extent of increasing in Cr.I is lower than the case of CMC, as a result of decreasing the carboxyl group content. The lowest content of Ag(I) in complex formation is based on COO$^-$ contents as chelating side.

The OH functional groups included the St provide weak ionic interaction of OH with Ag(I) ions, and consequently, it behaves in reverse trend (decrease in MHBS) than other biopolymers.

The proposed structure of SC–Ag-complexes formation versus CMC–Ag(I), OC–Ag(I) and St–Ag(I) complexes is as shown in figure 2.

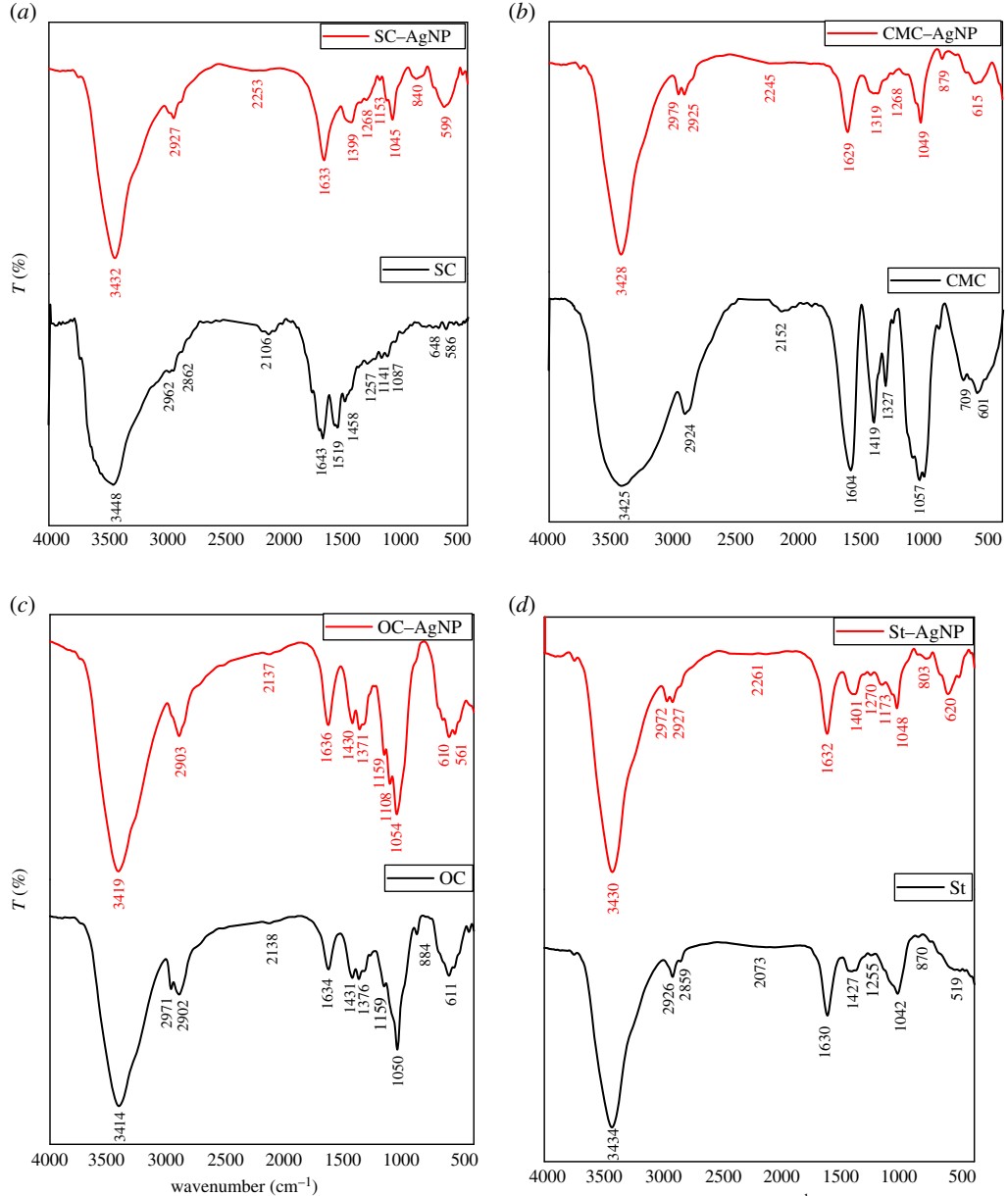

**Figure 1.** FTIR spectra of protein-based biopolymer versus cellulose- and starch-based biopolymers and their Ag-complexes.

**Table 1.** FTIR measurements of biopolymers and their complexes with Ag(I) ions.

| sample code | MHBS $(A(OH)_{st.}/A(CH)_{st.})$ | Cr.I. $(A_{\sim 1370\ cm^{-1}}/A_{\sim 2900\ cm^{-1}})$ |
|---|---|---|
| SC | 4.52 | 1.11 |
| SC–AgNP | 4.80 | 3.35 |
| CMC | 1.85 | 0.55 |
| CMC–AgNP | 6.04 | 2.13 |
| OC | 2.62 | 0.55 |
| OC–AgNP | 2.65 | 1.00 |
| St | 6.58 | 0.74 |
| St–AgNP | 5.40 | 0.83 |

**Figure 2.** Possibilities of Ag-complex formation versus biopolymer type. (R: remain of biopolymer unit).

### 3.1.2. Thermal stability

The change in the structure of Ag-complex as a result of biopolymer type and the chelating sites will lead to the occurrence of the variation in its thermal stability. In this respect, the non-isothermal thermogravimetric analysis of SC as protein-based biopolymers and its complex with Ag(I) ions (AgNPs) versus SCMC, OC and St as carbohydrate-based biopolymers complexes was studied. The TGA and DTGA curves and measurements are illustrated in figure 3 and table 2.

The thermal decomposition of biopolymers (SC, SCMC, OC and St) proceed in addition to the evolution of the sorbed moisture stage (less than 110°C), to the decomposition of functional groups and the main chain of biopolymers (depolymerization and oxidation) with the evolution of gases (volatilization stage), followed by carbonization stage and formation of carbonaceous char [40]. The formation of $Na_2CO_3$ is probable in the char in case of SC and CMC biopolymers.

TGA and DTGA of unchelated biopolymers showed different degradation behaviour, with respect to shape, onset temperature, temperature range and number of main degradation stages, DTG peak temperature, as well as weight remain (table 2). Where, the volatilization stage of sodium caseinate, which refers to its thermal stability, is achieved in two stages (at DTG peak temperature 154.3 and 319.6°C); while for other bioplymers (CMC, OC, St) only one stage is observed at DTG peak 281.2°C,

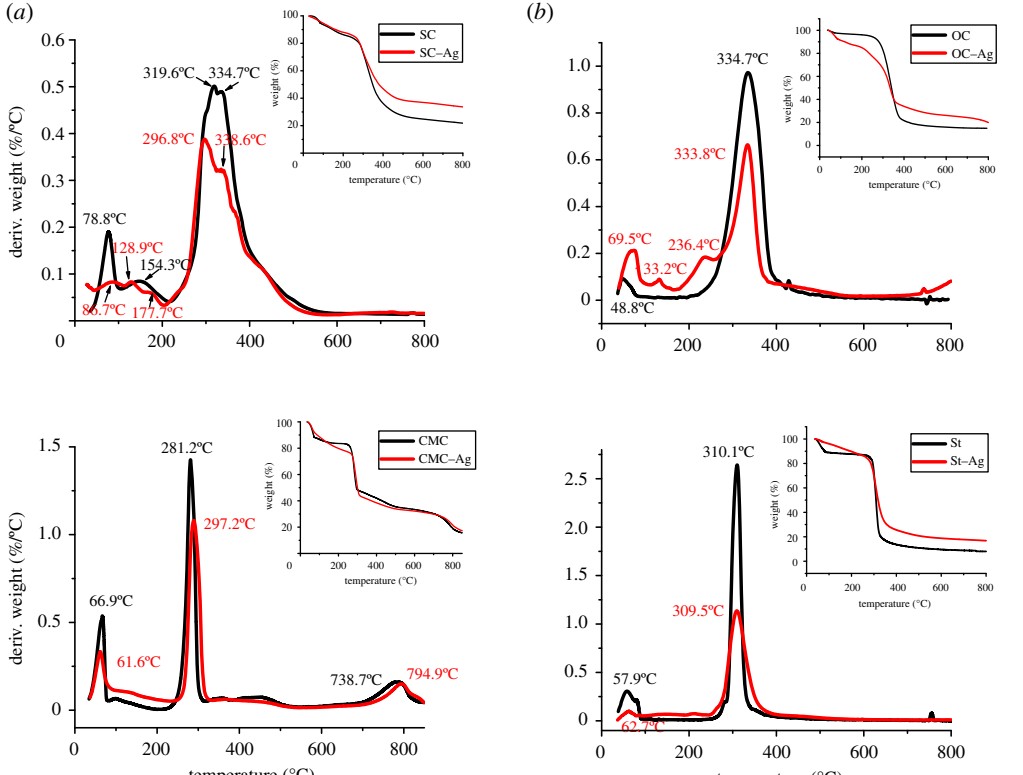

**Figure 3.** (*a*) TGA and DTGA non-isothermal curves of protein-based biopolymer versus cellulose-based biopolymer (CMC) and their Ag-complexes. (*b*) TGA and DTGA non-isothermal curves of protein-based biopolymer versus cellulose- and starch-based biopolymers and their Ag-complexes.

334.7°C and 310.1°C, respectively. The additional degradation stage in case of SC is probably related to the degradation of bonds at end groups [HOOC-R-NH] with volatilization of $CO_2$, $H_2O$, $NH_3$, HNCO and CO, then degradation of internal bonds of SC molecules, also with the evolution of $CO_2$ and $H_2O$. While, for cellulose derivatives (CMC and OC) and starch (St) based biopolymers, the decomposition proceeds via depolymerization with reduction in molecular weight by bond scission, the formation of free radicals, elimination of water, formation of carbonyl, carboxyl and hydroperoxide groups, the evolution of carbon monoxide and carbon dioxide, and finally production of a charred residue [40].

With regard to AgNPs produced from complexation of different biopolymers with Ag(I) ions, it is clear that (figure 3 and table 2), the SC–Ag-complex shows an additional peak at 128°C versus unchelated SC, with increasing the temperature of volatilization stage from 154 to 178°C. This additional peak together with the increasing temperature of the volatilization stage is confirmed by the formation of chelated bonds between Ag and SC via functional groups (COO or $NH_2$).

The onset temperatures and DTG peak temperatures of CMC–Ag are shifted from 264 to 269°C and from 281 to 297°C, respectively. As can be seen, the greatest increase in total activation energy of degradation (thermal stability) of carboxyl-included protein-based Ag-complex ($\Sigma E_a$ from 530 to 804.6 kJ mol$^{-1}$) than caboxyl-included cellulose–Ag complex ($\Sigma E_a$ from 675.09 to 829.86 kJ mol$^{-1}$). This is ascribed to bond formation between Ag(I) ions with the functional group including SC, via covalent coordinated or ionic bonds. The possible structure of SC–Ag complex with six-membered ring provided higher thermal stability than the case of five-member ring as the case of CMC–Ag complexes. In other words, and based on the greatest affinity of SC to chelate Ag(I) ions than CMC, as clear from the Ag contents (table 1) included SC–Ag (16.9 mM) and SCMC–Ag complexes (5.0 mM). This leads to the formation of more SC–Ag chelated bonds, which need higher energy for decomposition. The data of thermal stability or activation energy of degradation is the net of bond formation (coordinated and ionic bonds) and weakness of hydrogen bonds due to chelating of Ag(I) ions with $COO^-$ and $NH_2$ groups, as observed from data of MHBS (table 1).

The TGA and DTGA curves of oxidized cellulose as ligand for the formation of Ag-complex proceeded differently, where two additional peaks appeared together with the main volatilization stage in DTG peaks at 133, 236°C, with net activation energy of degradation reached 559 kJ mol$^{-1}$ instead of 241 kJ mol$^{-1}$ in case

**Table 2.** Non-isothermal TGA measurements of biopolymers and their complexes with Ag(I) ions.

| code | decomp. stage | temperature range (°C) ($T_O$–$T_E$) | DTG peak (°C) | weight loss (%) | weight loss | ash at 800°C | $n$ order | $R^2$ | s.e. | $E_a$ (kJ mol$^{-1}$) |
|---|---|---|---|---|---|---|---|---|---|---|
| SC | 1st | 35.20–104.2 | 78.8 | 6.73 | 7.11 | 21.8 | — | — | — | — |
| | 2nd | 109.3–212.0 | 154.3 | 7.2 | 13.12 | | 2.0 | 0.956 | 0.209 | 117.51 |
| | 3rd | 281.1–325.8 | 319.6 | 19.4 | 62.39 | | 2.0 | 0.945 | 0.202 | 413.50 |
| | 4th | 325.8–410.4 | 334.7 | 26.3 | | | 1.5 | 0.947 | 0.176 | 236.87 |
| | | | | | | | | | | $^a\Sigma E_a = 530$ |
| SC–Ag | 1st | 39.80–115.5 | 86.7 | 5.80 | 15.86 | 33.7 | — | — | — | — |
| | 2nd | 117.9–158.8 | 128.9 | 3.11 | | | 2.0 | 0.945 | 0.205 | 225.14 |
| | 3rd | 158.8–202.5 | 177.7 | 2.29 | | | 2.0 | 0.955 | 0.135 | 255.78 |
| | 4th | 264.7–323.3 | 296.8 | 18.9 | 54.64 | | 2.0 | 0.954 | 0.199 | 323.68 |
| | 5th | 330.1–424.3 | 338.6 | 19.4 | | | 2.0 | 0.921 | 0.245 | 245.24 |
| | | | | | | | | | | $\Sigma E_a = 804.6$ |
| CMC | 1st | 33.90–81.90 | 66.9 | 12.2 | 16.53 | 19.2 | — | — | — | — |
| | 2nd | 264.8–317.8 | 281.2 | 32.9 | 52.23 | | 1.5 | 0.982 | 0.156 | 447.84 |
| | 3rd | 378.5–525.9 | 456.5 | 8.49 | 22.78 | | 2.0 | 0.953 | 0.216 | 227.25 |
| | | | | | | | | | | $^a\Sigma E_a = 675.09$ |
| CMC–Ag | 1st | 40.80–90.30 | 61.7 | 10.1 | 23.00 | 21.7 | — | — | — | — |
| | 2nd | 269.6–317.2 | 289.4 | 18.7 | 47.48 | | 2.0 | 0.942 | 0.213 | 371.35 |
| | | | | | | | 2.5 | 0.941 | 0.264 | 458.51 |
| | 3rd | 742.4–844.9 | 794.9 | 10.5 | 20.92 | | 2.0 | 0.952 | 0.209 | 686.75 |
| | | | | | | | | | | $^*\Sigma E_a = 829.86$ |
| OC | 1st | 36.90–81.60 | 48.8 | 2.60 | 4.64 | 14.9 | — | — | — | — |
| | 2nd | 237.9–396.5 | 334.7 | 73.3 | 80.5 | | 2.5 | 0.976 | 0.264 | 241.28 |

(Continued.)

**Table 2.** (*Continued.*)

| code | decomp. stage | temperature range (°C) ($T_0$–$T_F$) | DTG peak (°C) | weight loss (%) | weight loss | ash at 800°C | n order | $R^2$ | s.e. | $E_a$ (kJ mol$^{-1}$) |
|---|---|---|---|---|---|---|---|---|---|---|
| OC–Ag | 1st | 40.90–90.30 | 69.5 | 8.50 | 13.62 | 19.9 | — | — | — | — |
| | 2nd | 110.5–152.7 | 133.2 | 2.99 | | | — | — | — | — |
| | 3rd | 179.9–256.7 | 236.4 | 10.2 | 86.38 | | 2.5 | 0.951 | 0.252 | 229.05 |
| | 4th | 285.9–380.0 | 333.8 | 35.3 | | | 2.5 | 0.969 | 0.233 | 329.97 |
| | | | | | | | | | | $\Sigma E_a = 559.02$ |
| St | 1st | 40.00–89.10 | 57.90 | 10.9 | 13.26 | 8.01 | — | — | — | — |
| | 2nd | 289.6–327.1 | 310.1 | 62.5 | 80.16 | | 2.0 | 0.959 | 0.248 | 694.38 |
| St–Ag | 1st | 36.50–81.60 | 62.70 | 3.13 | 13.40 | 16.8 | — | — | — | — |
| | 2nd | 276.6–353.6 | 309.5 | 52.5 | 72.96 | | 2.0 | 0.970 | 0.196 | 326.22 |

[a]Total activation of degradation related to volatilization stage.

of OC-based AgNPs. The appearance of peak at 133°C confirms the inclusion of coordinated water in complex molecule. Due to ionic bond formation between COO⁻ with Ag(I), additional peak was observed at 236°C which needs higher energy for bond cleavage of this bond.

On complexation of the starch with Ag ions, the trend is reversed, whereas decrease in onset temperature of the volatilization stage (from 289.6 to 276.6°C) was observed with decreasing activation energy for degradation from 694.4 to 326.2 kJ mol⁻¹. The explanation of this unacceptable observation is probably related to the net of bond formation from chelating the silver ions, where it provides in the formation of ionic bonds between OH groups with silver, and at the same time weakness the degree of hydrogen bonds and freeness the hydroxyl groups. This view was emphasized from the FTIR measurements (MHBS and Cr.I) (table 1).

It is evident that, due to the inclusion of AgO in char at the end of degradation stages, the greatest weight remained in case of Ag-complexes was more observed than in Ag-free biopolymers. The trend of increasing the char is in agreement with the amount of Ag content in complex which were determined by UV.

## 3.2. Transmission electron microscope (TEM)

This test was carried out to assess the role of protein-based biopolymer on the performance of Ag-complex as nanoparticles form (AgNPs), versus the other cabohydrate-based biopolymers (CMC, oxidized cellulose and starch). This will give some knowledge of dealing with the ability of the formed biopolymer–Ag to be used as biological active material for inhibiting the growth of microorganisms due to its ability to adhere to the cell wall of the microorganism, then penetrates and changes the permeability of cell membrane, followed by cell death. Moreover, the 'pits' are possible to be formed on the cell surface with the accumulation of the nanoparticles [18,19].

The TEM photographs of AgNPs from different biopolymers behave differently (figure 4). The smallest particle size is observed in case of SC and CMC more than OC and St. CMC provides homogeneous particle size (5.7–6.6 nm); while the particle size of SC is sited between 2.5 and 6.7 nm. For OC and St, values between 16–20 nm and 26 to approximately 40 nm were observed, respectively. The increase in the amount of AgNPs accumulation is clear in case of St-biopolymers; while OC-based AgNPs shows little nanoparticles on fibres. This is related to the data of Ag content in complex formation (table 1). The role of nano Ag–biopolymer complexes on their performance as biological active materials will be evaluated in further study (tables 3 and 4 and figure 5a–d).

## 3.3. Antimicrobial evaluation

With regard to the role of AgNPs produced from complexation of protein-based biopolymer versus CMC and other carbohydrates (OC and St), as antimicrobial agents, evaluation against pathogenic microorganisms was investigated.

Table 3 shows the diameter of the inhibition zone around the biopolymer–Ag complex. It is clear that AgNPs from complexation of SC with Ag(I) provide effective AgNPs against the examined microorganisms more than other biopolymer–Ag complexes, while the lowest inhibition was observed in case of OC-based AgNPs. The explanation of this observation was ascribed to both type of biopolymer and content of Ag(I) ions in the produced complex. The observed higher efficiency of these AgNPs towards bacteria than fungus is related to the fact that eukaryotic cells are more sensitive to heavy metals than prokaryotic cells. Therefore, the greatest Ag content in SC–Ag leads to the increase in the inhibition zone diameter. The same trend was noted in case of St–Ag-complex versus OC–Ag-complex.

Regarding the smallest concentration of synthesized nano silver compounds in the series (MIC) that prevents the development of visible growth of the test organism, table 4 shows that *S. aureus* (Gram-positive bacteria) needs relatively high amount of complexes for inhibition; while the lowest amount is observed in case of *Pseudomonas aeuroginosa*. Based on the foregoing data, SC–Ag-complex is recommended as the best AgNPs agent for inhibiting the pathogenic microorganisms and its behaviour is nearly the same as the reference antibiotic 'ciprofloxacin'.

### 3.3.1. Anti-tumour activity

This study was carried out *in vitro* against the human tumour cell line(s) of colon HCT116, lung carcinoma A549, human hepatocellular carcinoma HePG 2, prostate PC3 and breast cancer MCF7. These selected cancer cell lines are the most common cancer worldwide, for example, in Egypt the

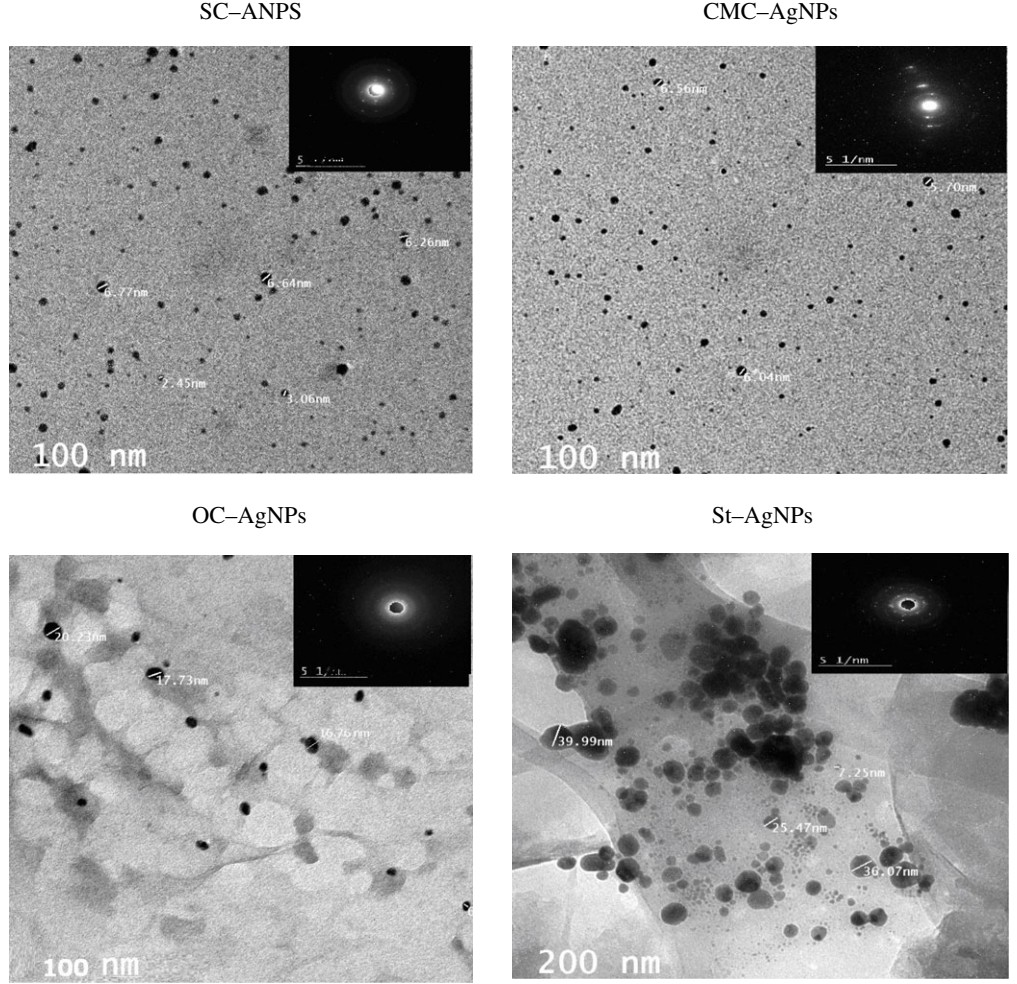

**Figure 4.** TEM of protein-based biopolymer versus cellulose- and starch-based biopolymers and their Ag-complexes.

**Table 3.** Antimicrobial activity by agar diffusion method of tested nano silver compounds versus biopolymer type. Highly active (+++) = (inhibition zone greater than or equal to 20 mm). Inactive (−ve) = no inhibit ion zone.

| | microorganism inhibition zone diameter (mm) | | | | |
| --- | --- | --- | --- | --- | --- |
| | Gram-positive bacteria | | Gram-negative bacteria | | fungi |
| compds. | *Bacillus subtilis* | *Staphylococcus aureus* | *Escherichia coli* | *Pseudomonas aeuroginosa* | *Candida albicans* |
| SC–AgNPs | 23 | 21 | 22 | 20 | 18 |
| CMC–AgNPs | 20 | 19 | 21 | 20 | 17 |
| OC–ANPs | 14 | 12 | 13 | 13 | 11 |
| St–AgNPs | 16 | 15 | 17 | 15 | 13 |
| ciprofloxacin | 19 | 20 | 23 | 21 | −ve |
| nystatin | −ve | −ve | −ve | −ve | 23 |

colon cancer affects both men and women aged 50 years or older [41]. The potential effect of the synthesized AgNPs from different protein- and carbohydrate-based biopolymers on inhibiting the cancer cell growth with decreasing the cell survival against concentration was clear in figure 5a,b.

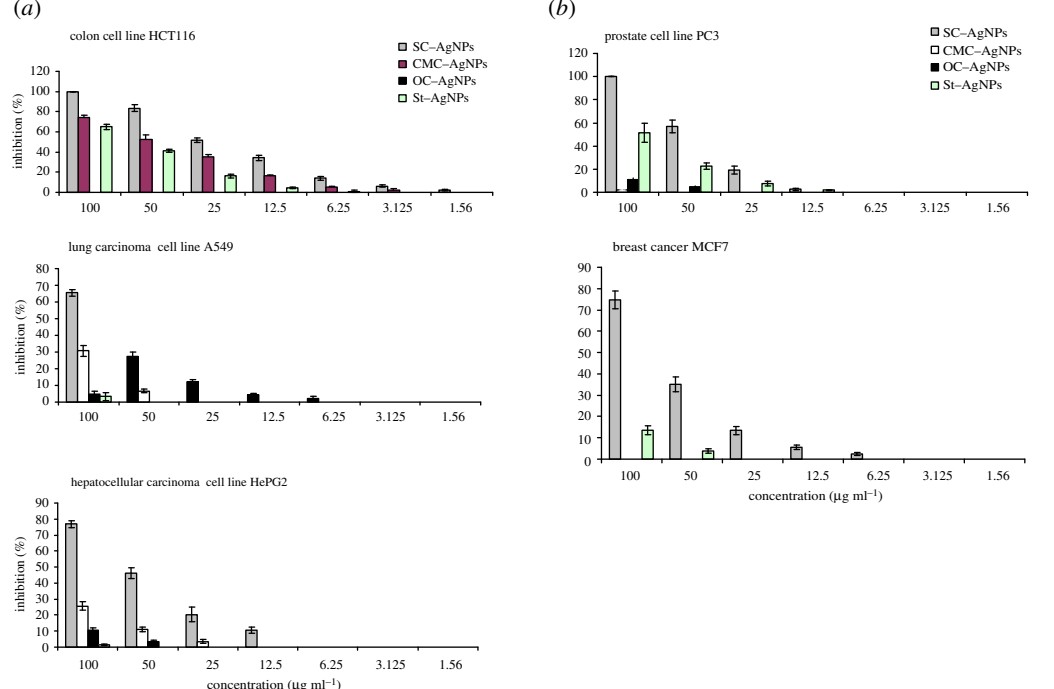

**Figure 5.** (*a*) Bioactivity of AgNPs from protein-based biopolymer versus carbohydrate-based biopolymers. (*b*) Bioactivity of AgNPs from protein-based biopolymer versus carbohydrate-based biopolymers.

**Table 4.** MIC of the synthesized nano silver particles ($\mu g\ ml^{-1}$) against tested microorganisms.

| synthesized nano silver samples | MIC of AgNPs ($\mu g\ ml^{-1}$) | | | | |
| --- | --- | --- | --- | --- | --- |
| | Gram-positive bacteria | | Gram-negative bacteria | | fungi |
| | *Bacillus subtilis* | *Staphylococcus aureus* | *Escherichia coli* | *Pseudomonas aeuroginosa* | *Candida albicans* |
| SC–AgNPs | 4 | 6 | 4 | 3 | 4 |
| CMC–AgNPs | 5 | 6 | 5 | 4 | 5 |
| OC–AgNPs | 7 | 8 | 7 | 6 | 7 |
| St–AgNPs | 6 | 7 | 6 | 5 | 6 |

The lethal concentrations of the AgNPs which cause the death of 50% and 90% of cells in 48 h were recorded. Moreover, we reported the selective index (SI) to detect the safety performance of examined biopolymers–AgNPs (table 5), which reflects the correlation between the cytotoxic activity against normal cell (RPE-1, telomerase immortalized normal human retinal epithelial cell line), and the cytotoxic activity against the tested cancer cells. The highest SI ratio (greater than 2) indicated the effectiveness and safety of the examined anti-cancer compound [42,43].

For all AgNPs, the dose 100 $\mu g\ ml^{-1}$ provides the greatest inhibition, and SC–AgNPs leads to 100% antiproliferative activity against colon and prostate PC3 cell lines. The SC–AgNPs was detected as significant anti-cancer compound for inhibition of cell proliferation of all examined cancer cell lines. Low effective dose of $IC_{50}$ and $IC_{90}$, especially towards colon cell line (25.8 and 54.73 $\mu g\ ml^{-1}$, respectively) was observed. The anti-cancer behaviour of CMC–AgNPs was only noted in inhibiting the colon cancer cell with doses greater than those of SC–AgNPs ($IC_{50}$ and $IC_{90}$ were 41.7 and 71.9 $\mu g\ ml^{-1}$, respectively). St–AgNPs also has effective role on inhibiting the growth of colon and prostate cell lines with high doses to perform 50% and 90% inhibition. Based on the value of SI, both SC and CMC-based AgNPs are recommended as safety anti-cancer treating agent; in comparison with St–AgNPs.

**Table 5.** Cytotoxic activity of AgNPs versus biopolymer-based AgNPs. $IC_{50}$: lethal concentration of the sample which causes the death of 50% of cells in 48 h. $IC_{90}$: lethal concentration of the sample which causes the death of 90% of cells in 48 h.

| cell line | sample code | $IC_{50}$ ($\mu$g ml$^{-1}$) | $IC_{90}$ ($\mu$g ml$^{-1}$) | RPE1 (normal retina cell line) | | |
| --- | --- | --- | --- | --- | --- | --- |
| | | | | $IC_{50}$ ($\mu$g ml$^{-1}$) | inhibition % at 100 ($\mu$g ml$^{-1}$) | SI |
| colon cell line HCT116 | SC–AgNPs | 25.8 | 54.73 | — | 22.4% | $\gg$3.9 |
| | CMC–AgNPs | 41.7 | 71.9 | — | 16.8 | $\gg$2.4 |
| | OC–AgNPs | — | — | — | 11.8 | — |
| | St–AgNPs | 63.5 | 101.1 | — | 45.3 | >1.6 |
| lung carcinoma cell line A549 | SC–AgNPs | 80.1 | 127.7 | — | 22.4% | 2.8 |
| | CMC–AgNPs | — | — | — | 16.8 | — |
| | OC–AgNPs | — | — | — | 11.8 | — |
| | St–AgNPs | — | — | — | 45.3 | — |
| human hepatocellular carcinoma cell line HePG2 | SC–AgNPs | 64.3 | 110.7 | — | 22.4% | 3.5 |
| | CMC–AgNPs | — | — | — | 16.8 | |
| | OC–AgNPs | — | — | — | 11.8 | |
| | St–AgNPs | — | — | — | 45.3 | |
| prostate cell line PC3 | SC–AgNPs | 45.1 | 66.7 | — | 22.4% | $\gg$2.2 |
| | CMC–AgNPs | — | — | — | 16.8 | — |
| | OC–AgNPs | — | — | — | 11.8 | — |
| | St–AgNPs | 92.9 | 144.0 | — | 45.3 | 1.2 |
| breast adenocarcinoma MCF7 | SC–AgNPs | 71.4 | 114.8 | — | 22.4% | 3.1 |
| | CMC–AgNPs | — | — | — | 16.8 | — |
| | OC–AgNPs | — | — | — | 11.8 | — |
| | St–AgNPs | — | — | — | 45.3 | — |
| | DMSO | —— | —— | 1 | | |
| | negative control | —— | —— | 0 | | |

The OC–AgNPs is not accepted for cancer treatment, where the maximum inhibition (21.5%) is noted in case of colon cell line.

The changes that occurred in treated cell lines as a function of biopolymer used in the production of AgNPs may be ascribed to the amount of Ag-chelated ions, particle size, as well as hydrophilic behaviour. The higher effect of SC-based AgNPs than the well-known CMC–AgNPs and other biopolymer-based AgNPs, in addition to its small particle size of 2.5–6.7 nm, is ascribed to the relatively high silver content, stability (from the values of MHBS and Cr.I) and hydrophilic behaviour (from the inclusion of hydroxyphosphate and unchelated amide), which enhance its affinity to penetrate the tumour cell and attack the functional protein, and consequently cause cell shrinkage via partial unfolding and aggregation of the protein, leading to restriction of the cancer cell growth pathway. While, in case of low stability particle size of AgNPs, as the case of St–AgNPs, the Ag ions are able to liberate and bind to the negatively charged cell surface with decreasing capability of shrinking and crossing the tumour cell.

## 4. Conclusion

Preparation of green Ag(I)-complex with nanostructure from protein origin with characterization as effective bioactive material (antimicrobial and anti-cancer), in comparison with that produced from the well-known used CMC and carbohydrate-biopolymers is the objective of this current work. The higher Ag content, MHBS and Cr.I together with the formation of six-membered chelate ring

geometrical structure of SC–Ag(I) complex, proved its superiority in biological application than five-membered chelate ring geometry of CMC–Ag complex, and linear structure of both OC–A(I) and St–Ag(I) complex. This promising complex provided the greatest inhibition zone against the tested microorganisms, and it was able to inhibit all examined cancer cell lines. With the most acceptable lethal concentration which causes the death of 50% colon, and breast cell lines than the literature, were reported pyridine derivative, and CMC-amino acid conjugates.

Ethics. This work was achieved by the researchers in National research Centre, using the facilities available in their laboratories.

Data accessibility. The datasets supporting this article have been uploaded as part of the supplementary material. It includes charts of TGA and FTIR of biopolymer and their resulting AgNPs. Data are available from the Dryad Digital Repository: https://doi.org/10.5061/dryad.7h44j0zrp [44].

Authors' contributions. All authors shared in the preparation of this article, the main contribution was achieved by A.H.B.; V.F.L shared in parts of the article dealing the synthesis of bioactive AgNPs, and evaluation; while the other authors (K.M.M. and N.A.M.A.) shared in parts of the biological and anti-cancer evaluation.

Competing interests. We declare we have no competing interests.

Funding. Funding is self-employed from all authors.

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
