## [Reviewer comments · Royal Society Open Science]

Review History

RSOS-200928.R0 (Original submission)

Review form: Reviewer 1

Is the manuscript scientifically sound in its present form?

No

Are the interpretations and conclusions justified by the results?

Yes

Is the language acceptable?

Yes

Do you have any ethical concerns with this paper?

Yes

Have you any concerns about statistical analyses in this paper?

No

Recommendation?

Accept with minor revision (please list in comments)

Comments to the Author(s)

Manuscript "Effectiveness of protein-based biopolymer in production of silver nanoparticles as high efficient bioactive compound versus Carbohydratesbased biopolymers" is looking valuable but acceptable after revision

Title: Dear author kindly revise title and make more simple and eye catchy for readers

Abstract: is too lengthy and silver material related information missing, kindly update with release kinetics and size and stability data.

Introduction:

Start from here "Many years ago, silver regards as the unique metal of valuable uses in treating and preventing the diseases. Its complexes were applied as biocide and/or biostatic [21]; while its salts succeeded in treating some infections, appearing resistance to the modern antibiotics and drops of silver nitrate are placed in newborn`s eyes at birth for the prevision of contracting gonorrhoea from mother [22]. Synthesizing of green silver nanoparticles gained attention by many researchers due to its biological activity, as a result of its ability to anchor the microorganism cell wall, followed by penetrating, changing the cell structure that can lead to cell [23, 24]. " and introduction part much be enriched by mechanistic approach of biological function of silver nano-particles.

Author may be improve introduction part in the light of following references--Antibacterial silver nanomaterials synthesis from

Obliteration of bacterial growth and biofilm through ROS generation by facilely synthesized green silver nanoparticles. PloS one 12 (8), e0181363

-Mesoflavibacter zeaxanthinifaciens and targeting biofilm formation. Frontiers in pharmacology 10, 801.

-Antimicrobial and anticancer activities of silver nanoparticles synthesized from the root hair extract of Phoenix dactylifera. Materials Science and Engineering: C 89, 429-443

Materials and Methods: performed well

Results and discussion: Good

Conclusion: kindly remove data values from conclusion

Review form: Reviewer 2

Is the manuscript scientifically sound in its present form?

Yes

Are the interpretations and conclusions justified by the results?

Yes

Is the language acceptable?

Yes

Do you have any ethical concerns with this paper?

No

Have you any concerns about statistical analyses in this paper?

Yes

Recommendation?

Major revision is needed (please make suggestions in comments)

Comments to the Author(s)

his work focuses on protein-based and carbohydrate-based biopolymer in production of silver nanoparticles, evaluating the antimicrobial activities and anti-tumor activities.

The novelty of this study relies on the comparison between protein-based-Ag nanoparticles and carbohydrate-based-Ag nanoparticles.

Overall this manuscript needs drastic revision before acceptance for publication.

- 1) When discussing the electiveness of protein-based biopolymer in production of silver nanoparticles, how to evaluate the effectiveness while there's not enough data to support it, such as yield and load of AgNPs?
- 2) Fig.2 shows the possibilities of Ag- complex formation vs biopolymer type, I'm afraid it's not proper to display the structure of biopolymers in your way without long polymer chain.
- 3) The connection mode between AgNPs and polymers isn't clear though FTIR assay can prove the bind with little difference.
- 4) In the assay of MIC, all samples were dissolved in DMSO. There's no doubt causing protein denaturation, making the results implausible.
- 5) The paper is riddled with grammatical or spelling mistakes, needing drastic revision. For example, FTIR-Spectra is spelt as FTRR-Spectra, biopolymers is spelt as biopoltmers and there's no dot in line 12 between 2106 cm⁻¹ and These bands.

Decision letter (RSOS-200928.R0)

Dear Professor Basta:

Title: Electiveness of protein-based biopolymer in production of silver nanoparticles as high efficient bioactive compound versus Carbohydrates-based biopol
 Manuscript ID: RSOS-200928

The editor assigned to your manuscript has now received comments from reviewers. We would like you to revise your paper in accordance with the referee and Subject Editor suggestions which can be found below (not including confidential reports to the Editor). Please note this decision does not guarantee eventual acceptance.

Please submit your revised paper before 02-Sep-2020. Please note that the revision deadline will expire at 00.00am on this date. If we do not hear from you within this time then it will be assumed that the paper has been withdrawn. In exceptional circumstances, extensions may be possible if agreed with the Editorial Office in advance. We do not allow multiple rounds of revision so we urge you to make every effort to fully address all of the comments at this stage. If deemed necessary by the Editors, your manuscript will be sent back to one or more of the original reviewers for assessment. If the original reviewers are not available we may invite new reviewers.

To revise your manuscript, log into <http://mc.manuscriptcentral.com/rsos> and enter your Author Centre, where you will find your manuscript title listed under "Manuscripts with Decisions." Under "Actions," click on "Create a Revision." Your manuscript number has been

appended to denote a revision. Revise your manuscript and upload a new version through your Author Centre.

RSC Associate Editor:
Comments to the Author:
(There are no comments.)

RSC Subject Editor:
Comments to the Author:
(There are no comments.)

Reviewers' Comments to Author:
Reviewer: 1

Comments to the Author(s)
Manuscript "Electiveness of protein-based biopolymer in production of silver nanoparticles as high efficient bioactive compound versus Carbohydratesbased biopolymers" is looking valuable but acceptable after revision

Title: Dear author kindly revise title and make more simple and eye catchy for readers
Abstract: is too lengthy and silver material related information missing, kindly update with release kinetics and size and stability data.

Introduction:
Start from here "Many years ago, silver regards as the unique metal of valuable uses in treating and preventing the diseases. Its complexes were applied as biocide and/or biostatic [21]; while its salts succeeded in treating some infections, appearing resistance to the modern antibiotics and drops of silver nitrate are placed in newborn's eyes at birth for the prevision of contracting

gonorrhoea from mother [22]. Synthesizing of green silver nanoparticles gained attention by many researchers due to its biological activity, as a result of its ability to anchor the microorganism cell wall, followed by penetrating, changing the cell structure that can lead to cell [23, 24]. " and introduction part much be enriched by mechanistic approach of biological function of silver nano-particles.

Author may be improve introduction part in the light of following references-

-Antibacterial silver nanomaterials synthesis from

Obliteration of bacterial growth and biofilm through ROS generation by facilely synthesized green silver nanoparticles. PLoS one 12 (8), e0181363

-Mesoflavibacter zeaxanthinifaciens and targeting biofilm formation. Frontiers in pharmacology 10, 801.

-Antimicrobial and anticancer activities of silver nanoparticles synthesized from the root hair extract of Phoenix dactylifera. Materials Science and Engineering: C 89, 429-443

Materials and Methods: performed well

Results and discussion: Good

Conclusion: kindly remove data values from conclusion

Reviewer: 2

Comments to the Author(s)

his work focuses on protein-based and carbohydrate-based biopolymer in production of silver nanoparticles, evaluating the antimicrobial activities and anti-tumor activities.

The novelty of this study relies on the comparison between protein-based-Ag nanoparticles and carbohydrate-based-Ag nanoparticles.

Overall this manuscript needs drastic revision before acceptance for publication.

- 1) When discussing the electiveness of protein-based biopolymer in production of silver nanoparticles, how to evaluate the effectiveness while there's not enough data to support it, such as yield and load of AgNPs?
- 2) Fig.2 shows the possibilities of Ag- complex formation vs biopolymer type, I'm afraid it's not proper to display the structure of biopolymers in your way without long polymer chain.
- 3) The connection mode between AgNPs and polymers isn't clear though FTIR assay can prove the bind with little difference.
- 4) In the assay of MIC, all samples were dissolved in DMSO. There's no doubt causing protein denaturation, making the results implausible.
- 5) The paper is riddled with grammatical or spelling mistakes, needing drastic revision. For example, FTIR-Spectra is spelt as FTIR-Spectra, biopolymers is spelt as biopolymers and there's no dot in line 12 between 2106 cm⁻¹ and These bands.

Author's Response to Decision Letter for (RSOS-200928.R0)

See Appendix A.

RSOS-200928.R1 (Revision)

Review form: Reviewer 2

Is the manuscript scientifically sound in its present form?

Yes

Are the interpretations and conclusions justified by the results?

Yes

Is the language acceptable?

Yes

Do you have any ethical concerns with this paper?

No

Have you any concerns about statistical analyses in this paper?

No

Recommendation?

Accept as is

Comments to the Author(s)

The authors have carefully addressed the concerns of reviewers.

Decision letter (RSOS-200928.R1)

Dear Professor Basta:

Title: Synthesis and evaluation of protein-based biopolymer in production of silver nanoparticles as bioactive compound versus carbohydrates-based biopolymer

Manuscript ID: RSOS-200928.R1

It is a pleasure to accept your manuscript in its current form for publication in Royal Society Open Science. The chemistry content of Royal Society Open Science is published in collaboration with the Royal Society of Chemistry.

RSC Associate Editor:
Comments to the Author:
(There are no comments.)

RSC Subject Editor:
Comments to the Author:
(There are no comments.)

Reviewer(s)' Comments to Author:
Reviewer: 2

Comments to the Author(s)
The authors have carefully addressed the concerns of reviewers.

Appendix A

Dear Editor Prof. Anthony Stace and the Asso. Editor Dr Ya-Wen Wang.

First, we hope you are keeping well at this difficult period, too. May our LORD GOD safe you and your dears.

We would like to present my deep thanks for efforts made by you and reviewers to improve my submitted Ms with Ref. # (Ms ID: RSOS-200928). All the reviewers' remarks are considered in revised version, and the changes made in text are highlighted in grey color. Following it our Answers to Reviewer's remarks, and we hope this revised version will meet with approval.

Reviewer 1

* Manuscript "Electiveness of protein-based biopolymer in production of silver nanoparticles as high efficient bioactive compound versus Carbohydratesbased biopolymers" is looking valuable but acceptable after revision

Answer

Many thanks for your kind decision, which will give us the chance for extending our publication in this estimated Journal

* Title: Dear author kindly revise title and make more simple and eye catchy for readers

Answer

Thanks for your kind advice, the title was changed to " Synthesis and evaluation of protein-based biopolymer in production of silver nanoparticles as bioactive compound versus carbohydrates-based biopolymers".

* Abstract: is too lengthy and silver material related information missing, kindly update with release kinetics and size and stability data.

Answer

This section was modified and the # of words it reduced to 222, , with thanks

* Introduction

Start from here "Many years ago, silver regards as the unique metal of valuable uses in treating and preventing the diseases. Its complexes were applied as biocide and/or biostatic [21]; while its salts succeeded in treating some infections, appearing resistance to the modern antibiotics and drops of silver nitrate are placed in newborn`s eyes at birth for the prevision of contracting gonorrhoea from mother [22]. Synthesizing of green silver nanoparticles gained attention by many researchers due to its biological activity, as a result of its ability to anchor the microorganism cell wall, followed by penetrating, changing the cell structure that can lead to cell [23, 24]. " and introduction part much be enriched by mechanistic approach of biological function of silver nano-particles.

Author may be improve introduction part in the light of following references-

- Antibacterial silver nanomaterials synthesis from Obliteration of bacterial growth and biofilm through ROS generation by synthesized green silver nanoparticles. PloS one 12 (8), e0181363
- Mesoflavibacter zeaxanthinifaciens and targeting biofilm formation. Frontiers in pharmacology 10, 801.
- Antimicrobial and anticancer activities of silver nanoparticles synthesized from the root hair extract of Phoenix dactylifera. Materials Science and Engineering: C 89, 429-443

Answer

We added your valuable suggested References. Could you please check the highlighted 2nd paragraph in revised version, with thanks.

The 2nd paragraph of Introduction section of original article was deleted, and based on your advice we modified the 3rd paragraph and become to be 2nd. Based on this modification the References numbers were changed, and added the following References.

Dr. Please permit me to keep the 1st paragraph as it summarized the applications of biopolymers and their complexes with metal ions, with many thanks..

21. S. Qayyum, M. Oves, A. U. Khan, Obliteration of bacterial growth and biofilm through ROS generation by facilely synthesized green silver nanoparticles. PLoS one 12 (8), 2017, e0181363
22. M. Oves, M. Aslam, M. AhmarRauf, S. Qayyum, H. A. Qari, M. S. Khan, M. Z. Alam, S. Tabrez, A. Pugazhendhi, I. M. I. Ismail, Antimicrobial and anticancer activities of silver nanoparticles synthesized from the root hair extract of *Phoenix dactylifera*. Mater. Sci. and Eng. C, 89, 2018, 429-443
23. M. Oves, M. A. Rauf, A. Hussain, H. A. Qari, A. A. P. Khan, P. Muhammad, M. T. Rehman, M. F. Alajmi, I. I. M. Ismail, Antibacterial Silver Nanomaterial Synthesis From *Mesoflavibacter zeaxanthinifaciens* and Targeting Biofilm Formation. Frontiers in pharmacology 10, 2019, 801.
24. N. Durán, M. Durán, M. B. de Jesus, A. B. Seabra, W. J. Fávaro, G. Nakazato, Silver nanoparticles: A new view on mechanistic aspects on antimicrobial activity. Nanomed. Nanotechnol., Biol, Med., 12, 2016, 789–799

* Materials and Methods: performed well

* Results and discussion: Good

* Conclusion: kindly remove data values from conclusion

Answer

Once again thank you, I modified the conclusions as your advice.

-Reviewer 2

This work focuses on protein-based and carbohydrate-based biopolymer in production of silver nanoparticles, evaluating the antimicrobial activities and anti-tumor activities. The novelty of this study relies on the comparison between protein-based-Ag nanoparticles and carbohydrate-based-Ag nanoparticles.

Overall this manuscript needs drastic revision before acceptance for publication.

1) When discussing the electiveness of protein-based biopolymer in production of silver nanoparticles, how to evaluate the effectiveness while there's not enough data to support it, such as yield and load of AgNPs?

Answer

Thank you for your advice; also as response to the 1st Reviewer we changed the title to become "" Synthesis and evaluation of protein-based biopolymer in production of silver nanoparticles as bioactive compound versus carbohydrates-based biopolymers".

I hope it convenient with your inquiry.

2) Fig.2 shows the possibilities of Ag- complex formation vs biopolymer type, I'm afraid it's not proper to display the structure of biopolymers in your way without long polymer chain.

Answer

Thank you, We modified the proposed structures of biopolemer-based AgNPs in Fig. 2, as follow; I hope it will be convenient

Fig. 2: Possibilities of Ag-complex formation vs biopolymer type. (R: remain of biopolymer unit)

3) The connection mode between AgNPs and polymers isn't clear though FTIR assay can prove the bind with little difference.

Answer

Thanks for this valuable comment, but I ask you to check the highlighted paragraphs related to FT-IR spectra

- Under sub title 3.1.1, lines # 10-18, as we reported all changes accompanied the complexation of SC (as example) with Ag(I) ions, from shifting the bands related to NH and COO groups and the increase of band intensity at wave-number range related to M-O (Ag-O bond formation). Moreover, we realized the increase in Ag(I) content which was noticed in Table 1, with the proposed structure (presence more chelating gps.; NH and COO groups chelating sides),

- In case of CMC-AgNPs (in the same sub-title, 2nd paragraph), we proposed the chelating side with Ag(I) is via COO groups and not OH, based on the position of OH stretching vibration of OH band on complexation is slightly shifted to higher wave-number (from 3425 to 3428 cm^{-1}); while the bands related to C=O of COO⁻ are decreased with disappearance of band related to C-O (1419 cm^{-1}). This observation is evidence the OH groups not included the chelation, but the chelation sides are via COO⁻ and ether linkage of 1ry alcohol, with formation five member ring chelation geometry (Fig. 2).
- Also, these proposed structures were emphasized from the data of TGA measurements [please check the following paragraph which was reported in text of original version [highlighted paragraph in grey color, page # 9-10, started from the last line in page 9]

[The possible structure of SC-Ag complex with 6-membered ring is provided higher thermal stability than the case of five-member ring as the case of CMC-Ag complexes. In other words, and based on the greatest affinity of SC to chelate Ag(I) ions than CMC, as clear from the Ag contents (Table 1) included SC-Ag (16.9 mM) and SCMC-Ag complexes(5.0 mM). This leads to the formation of more SC-Ag chelated bonds, which need higher energy for decomposition. The data of thermal stability or activation energy of degradation is the net of bond formation (coordinated and ionic bonds) and weakness of hydrogen bonds due to chelating of Ag (I) ions with COO⁻ and NH₂ groups, as observed from data of MHBS (Table 1)].

- 4) In the assay of MIC, all samples were dissolved in DMSO. There's no doubt causing protein denaturation, making the results implausible.

Answer

Thank you for this comment, but we would like to clear, this test was carried out according to Ref. [36]. The use of DMSO (2% in Aqueous solution), in MIC, because it was the well known organic solvent that do not exhibit any inhibition on the pathogenic microorganism growth, beside the tested samples were found soluble in DMSO, but not soluble in other organic solvents. Following some published articles on using DMSO-antimicrobial test of AgNPs

- Wypij et al., Synthesis, characterization and evaluation of antimicrobial and cytotoxic activities of biogenic silver nanoparticles synthesized from *Streptomyces xinghaiensis* OF1 strain, *World Journal of Microbiology and Biotechnology* 34 (23), 2018, 1-13.
- Rodríguez-Luis, et al., Green Synthesis of Silver Nanoparticles and Their Bactericidal and Antimycotic Activities against Oral Microbes, *Journal of Nanomaterials*, Vol 2016, Article ID 9204573, 10 pages
- Aldabaldetrecu et al., Stability of Antibacterial Silver Carboxylate Complexes against *Staphylococcus epidermidis* and Their Cytotoxic Effects. *Molecules* 23, 2018, 1629" ;
- Ejidike, I.P. Cu(II) Complexes of 4-[(1E)-N-{2-[(Z)-Benzylidene amino]ethyl}ethanimidoyl]benzene-1,3-diol Schiff Base: Synthesis, Spectroscopic, In-Vitro Antioxidant, Antifungal and Antibacterial Studies. *Molecules* 23, 2018, 1581.

- 5) The paper is riddled with grammatical or spelling mistakes, needing drastic revision. For example, FTIR-Spectra is spelt as FTIR-Spectra, biopolymers is spelt as biopolymers and there's no dot in line 12 between 2106 cm^{-1} and These bands.

Answer

I'm extremely sorry about it. We checked the article and made the corrections about typo and language. All the changes are highlighted in grey color, once again thanks for your efforts and time.

Finally, I hope our response to Reviewers comments and the revised version are convenient for publication.

Kind regards,
Altaf H. Basta (Res. Prof.)
NRC, Cairo, Egypt